# Remote working and experiential wellbeing: A latent lifestyle perspective using UK time use survey before and during COVID-19

**Jerry Chen** , **Li Wan** *

Department of Land Economy, University of Cambridge, Cambridge, United Kingdom

* lw423@cam.ac.uk

## Abstract

Mental health in the UK had deteriorated compared with pre-pandemic trends. Existing studies on heterogenous wellbeing changes associated COVID-19 tend to segment population based on isolated socio-economic and demographic indicators, notably gender, income and ethnicity, while a more holistic and contextual understanding of such heterogeneity among the workforce seems lacking. This study addresses this gap by 1) combining UK time use surveys collected before and during COVID-19, 2) identifying latent lifestyles within three working mode groups (commuter, homeworker and hybrid worker) using latent class model, and 3) quantifying nuanced experiential wellbeing (ExWB) changes across workers of distinct lifestyles. The direction and magnitude of ExWB changes were not uniform across activity types, time of day, and lifestyles. The direction of ExWB change during the daytime activities window varied in accordance with lifestyle classifications. Specifically, ExWB decreased for all homeworkers but increased significantly for certain hybrid workers. Magnitude of ExWB change correlated strongly with lifestyle. To understand the significant heterogeneity in ExWB outcomes, a spatial-temporal conceptualisation of working flexibility is developed to explicate the strong yet complex correlations between wellbeing and lifestyles. The implications to post-pandemic "back-to-work" policies are 1) continued expansion of hybrid working optionality, 2) provide wider support for lifestyle adaptation and transitions.

## Introduction

Mental health in the UK had deteriorated compared with pre-pandemic trends [1]. Evidence of association between subjective wellbeing and the COVID-19 pandemic largely focused on the increase in psychological distress following emergency lockdown measures [2, 3]. While post-pandemic recovery progressed across the globe, prolonged social distancing and travel restrictions have led to profound societal changes [4, 5]–particularly concerning the future of work. In the UK, rapid democratisation of remote working has been instrumental in maintaining productivity of firms during the pandemic and flexible/remote working is likely to remain in practice for some occupations and sectors. Home-based and hybrid working is not new; however, the wellbeing changes associated with flexible/remote working is not fully

underlying the results presented in the study are directly accessible via the DOIs below: http://doi.org/10.5255/UKDA-SN-8128-1 http://doi.org/10.5255/UKDA-SN-8741-3

**Funding:** This work was supported by The Ove Arup Foundation for the Digital Cities for Change Programme research grant (RG89525 to LW), the EPSRC and UK Department for Transport joint grant for the Research Hub for Decarbonised Adaptable and Resilient Transport Infrastructures (DARe) (EP/Y024257/1 to LW), and the Cambridge Centre for Smart Infrastructure and Construction, which is funded by Innovate UK and EPSRC grants (EP/N021614/1, EP/I019308/1 and EP/K000314/1 to LW). JW is the recipient of the Stamps Scholarship hosted at Queens' College, University of Cambridge.

**Competing interests:** The authors have declared that no competing interests exist.

understood yet further complicated by our lived experience of the pandemic. Existing studies on subjective wellbeing during COVID-19 tend to segment population based on single socio-economic and demographic indicators, notably gender, income and ethnicity, while a more holistic, evolving understanding of heterogeneity among the workforce seems lacking.

Time-use data provides a unique source of information for understanding nuanced and evolving wellbeing during the COVID-19 pandemic [6–9]. This paper aims to address the above research gap by empirically identifying heterogeneous linkages between subjective wellbeing and flexible/remote working across three dimensions, 1) aggregate daily activity types, 2) distinct lifestyles of workers and 3) the evolution from pre- to during the pandemic. While the first dimension on activity types has been explored in the literature [9], the lifestyle perspective and the temporal dimension combined are new, thus contribute to our understanding of the association between flexible/remote working and wellbeing. In terms of analytical strategy, latent class model is applied to identify nuanced lifestyles within each of the three generic working mode groups (homeworkers, commuters, and hybrid workers). The novel use of latent class model on time-use data allows efficient and holistic segmentation of workers based on their socio-economic and demographic profiles and daily activity patterns. The temporal dimension is enabled by combining population-representative, repeated cross-sectional time use data gathered during the pandemic (2020) with the pre-pandemic UK Time Use Survey (UKTUS) in 2015.

Based on findings from the lifestyle perspective, we further propose a novel two-dimensional framework for conceptualising 'flexibility' as per flexible working in an intraday setting, which helps explicate strong yet complex correlations between wellbeing and daily activity pattern. The new framework builds on the work of Anttila et al. [10], featuring a separation of spatial and temporal dimension for measuring intraday working flexibility based on time-use data. *Temporal flexibility is often overlooked as spatial flexibility dominates existing discourse on flexible working practices.* The incorporation of the temporal dimension in conjunction with the lifestyle perspective is expected to shed light on the relationship between spatial-temporal working arrangements and subjective wellbeing. Implications for 'back-to-work' policies are discussed, emphasising the policy need for facilitating lifestyle transitions, as opposed to merely reverting the location of work.

## Literature review

### Measurements of subjective wellbeing

Leveraging the strength of the combined time-use data, subjective wellbeing is conceptualised as experienced or experiential wellbeing (ExWB). ExWB, sometimes referred to as hedonic wellbeing, is the series of momentary affective states that occur at one or a series of point-in-time reference periods [7]. Typical questions measuring ExWB are "How do you feel at this moment?" or "How did you enjoy commuting to work yesterday morning?". In contrast with evaluative wellbeing, which involves comparatively long periods and focuses on global assessments on one's state of being or satisfaction through e.g. the 12-item General Health Questionnaire (GHQ-12), ExWB measurement aims to capture a respondent's immediate focus, rather than broader issues that fall in the domain of evaluative wellbeing [7].

Compared to evaluative wellbeing, ExWB has been the lesser-studied aspect of subjective wellbeing. The reasons are twofold. Firstly, the point-in-time reference period likens ExWB to a fleeting emotional state, thus have questioned the volatility and clinical relevance of ExWB [11]. However, the temporal aspect is a unique strength of ExWB, as it captures emotions as they fluctuate from moment to moment and in response to day-to-day events and activities [8]. Secondly, despite the ability to capture instantaneous wellbeing, collecting ExWB data is

resource intensive [12]. For the measurement to have explanatory power, measurements associated with different time points and different activities throughout time need to be gathered. This requires a herculean effort of data collection and poses a large burden for the respondents [13]. Our research takes advantage of the high-granularity and population-representative ExWB data in the UKTUS data. The wealth of ExWB data available both pre and during the pandemic is a novel enabler for our research.

## Why are we focusing on time use?

During the pandemic, significant changes in both quantity and quality of time spent on different activities have been observed. Individuals spent more time on housework and less time on paid work activities during COVID-19 [9]. The paper also compared the first and third UK lockdowns with the pre-pandemic trend. They found that in the third lockdown, one fifth of the samples experienced increased atypical working hours, which strongly correlates with decreases in their ExWB. The disutility associated with increased atypical paid work hours is compounded by unenjoyable substitution activities during typical paid work hours, such as non-paid work (e.g. childcare) or alone leisure time. The authors called for further research on the factors that affects ExWB beyond the specific activity conducted.

Lee [14] focused on older adults over the age of 65 in the Korean Time Use Survey. Latent Class Analysis was used to identify four activity profiles across two waves in 2014 and 2019. The author aggregates daily activities into eight domains and summed up the time spent in each of these domains. The resulting eight continuous time measurements are then transformed into three-tier ordinal variables, which are used as the basis of classification. This approach focuses on the duration of activities, but ignores the timing and hence sequence of activities. As such, the study is not able to distinguish temporal patterns and sequences in its latent class assignments.

Adam et al. [15] leveraged the UKTUS 2015 data to study experiential wellbeing levels during commuting episodes. Averaging across all samples, time spent in passive commuting and paid work are associated with the lowest level of subjective wellbeing. The authors compared commuting days to non-commuting days, and found that after controlling for usual working arrangement and other socioeconomic characteristics, generally non-commuting days tend to lead to higher reported ExWB levels. The findings are consistent with an earlier study by the Office for National Statistics [16]. However, the authors found that notably there were no meaningful differences in ExWB levels for paid work activities between commuting and non-commuting days prior to the pandemic.

Zhou & Kan [17] analysed subjective wellbeing levels during three lockdown periods and the intermedial easing periods. Subjective wellbeing is measured through the GHQ-12 from the UK Household Longitudinal Survey COVID studies. The authors focused on the change in distress levels throughout the one-year period between April 2020 and March 2021. While time use patterns became less sensitive to the later lockdowns, distress levels reached a new high with repeated lockdowns in multiple waves of the pandemic. These negative wellbeing changes are heterogeneous across social groups. Subjective wellbeing decreased more rapidly for women, whereas men experienced the deterioration in wellbeing more gradually, but reached progressively higher levels of distress with each subsequent lockdown. Women recovered their wellbeing faster, and maintained a similar peak level of distress in all three lockdowns. BAME (Black, Asian, Minority Ethnic) groups experienced greater distress in the first lockdown.

### An evolving definition of flexible working

It is well researched that flexible working practices such as remote working enhances the sense of autonomy, job satisfaction, and other evaluative wellbeing measures [18–20]. However, COVID-19 profoundly changed the definition of flexible working. A systematic review on the relationship between remote work and productivity found stark contrast where before the pandemic 79% studies showed positive effects of remote work, that figure dropped to 23% during the pandemic [21]. Preliminary evidence show that remote workers experience more anxiety around job security [22], and are more prone to loneliness [23]. Cell-phone-based mobility data reveal trips to the CBD stabilising at 60% of pre-pandemic levels, suggesting remote workers will make up a significant portion of the future workforce [24].

Previous studies on the relationship between COVID-19 and workers' wellbeing are limited to specific activities (e.g. paid work in Barrero, Bloom and Davis [25]), or specific demographic groups (e.g. young adults in Gagne, Nandi and Schoon [26]). Few studies explicitly compared ExWB changes of distinct worker groups such as commuters, homeworkers, and hybrid workers. Cross-sectional data during the first lockdown in Portugal showed that homeworkers had a higher level of job satisfaction, due to better work-life balance and flexibility [27]. Among homeworkers there were yet more distinct groups with varied wellbeing outcomes. A multinational European study showed occasional homeworkers had high levels of satisfaction with their job quality, while highly mobile homeworkers reported poor work-life balance [28]. There was also evidence suggesting heterogeneous pandemic effects on mental health based on work environments [29].

A systematic review of ten controlled before and after studies found that increasing worker control and choice through flexible work interventions were likely to have a positive effect on health outcomes, but there was a clear need to delineate the complex relationship between flexible working and wellbeing [30]. National lockdowns in the UK were widely imposed across demographic and socioeconomic groups, however, studies involving occupational cohorts have largely been restricted to healthcare and or essential workers [31]. There is a clear gap in literature for a study that looks at the entire working population, to compare across population subgroups with different working arrangements, and to account for latent yet distinct lifestyles within these subgroups. Lifestyle differences and the shift to more flexible working practices enabled by remote work need to be accounted for to delineate the wellbeing changes associated with COVID-19. Through combining the pre- and during pandemic Time Use Survey data, this study aims to quantify the changing relationship between time use and wellbeing across heterogenous yet latent groups of employed workers in the UK.

## Data and methods

### Research ethics approval

Ethics approval for this study was not required. All data involved is openly available to the public before the initiation of the study.

### Data source

UKTUS (2015) is a large-scale household survey that provides data on how people spend their time. Time diaries record activity sequences, corresponding locations, as well as the respondent's level of subjective wellbeing throughout the day [32]. We combined the pre-pandemic UKTUS 2015 with four additional waves of population-representative time use data collected during the pandemic, which is a novel in time-use literature. These waves were collected online in 1) 2016; 2) May-June 2020 at the peak of first UK COVID-19 lockdown; 3) August 2020

following the relaxation of social restrictions 4) November 2020 during the second lockdown [33]. The latter three waves collected throughout 2020 were classified in our analysis as 'during COVID', while the 'pre-COVID' data consisted of UKTUS 2015 and the first online wave collected in 2016. Data was obtained from the UK Data Service, all released data waves were included in our analysis without selection or omission.

The study samples comprised all those aged 16–64, in full-time employment, with sufficient data on activity-level episodic ExWB, and a non-zero survey weight. We focus on the full-time employed as distinct lifestyles emerge, particularly during the pandemic, in this group of workers conventionally considered to follow homogenous working patterns. Our analysis leverages the unique strength of UKTUS and its 4-wave extensions during COVID-19, which include 3,855 individuals, each contributing 24 hours of time use schedule during a typical weekday in 10-minute episodes. A total of 369,394 activity-location-ExWB bundles were analysed (see sample distribution in *S1 Appendix*).

## Measurements

Prior to the identification of lifestyles using the latent class models, we first separated all samples by the reported location of paid work into three distinct working modes: 1) homeworkers; 2) commuters; 3) hybrid workers. Homeworkers engaged in paid work exclusively at home during the diary day; commuters engaged in paid work exclusively at the workplace; whilst hybrid workers reported paid work activities both at home and in the workplace. Such information had been collected consistently across survey periods. Other sociodemographic characteristics considered were income, educational attainment, occupation classification, gender, age, and marital status.

UKTUS has 144 distinct activity categories. To reduce the complexity, we followed literature [9] and aggregated them into broad activity types: 1) personal; 2) paid work; 3) non-paid work; 4) leisure; with our addition of 5) transport (see breakdown and aggregation methods in *S2 Appendix*. Intraday time use pattern was also aggregated from 10-minute intervals into 24 hourly variables, based on the broad activity type a respondent was engaged in for the majority of a given hour. Recognising the rigidity of certain working arrangement (e.g. the eight-hour working day that has largely persisted since the industrial era) and the need to capture lifestyle differences before and during COVID-19, the dairy day for each respondent is bifurcated into a "daytime activities window" (between 6am and 6pm) and a "evening activities window" (before 6am and after 6pm). The 12 consecutive categorical hourly variables thus fall into each observational window in a mutually exclusive manner, with 6am and 6pm as the boundaries. The rationale for these temporal boundaries is based on empirical observations (see details in the Results section).

A unique strength of UKTUS is the *enjoyment* measurement for every activity episode recorded. This paper conceptualises the enjoyment variable as a measurement of experiential wellbeing (ExWB), which assesses peoples' short-term emotions [34]. We recognise the limitation of using a single-measure wellbeing, but the selection of subjective wellbeing measures represent a difficult trade-off. While a multi-dimensional wellbeing measure (e.g. GHQ-12) would be more comprehensive and comparable across countries, the ExWB (enjoyment) measurement in UKTUS provides a high level of granularity that captures the intraday fluctuations of wellbeing, which is unique and worth further investigation. Arguably, the ExWB data in UKTUS seems to have been underutilised in literature given its uniqueness and granularity.

For every activity, a respondent's ExWB on the 7-point Likert scale is obtained, 1 represents "Did not enjoy at all", 2 represents "Did not enjoy", 3 represents "Slightly did not enjoy", 4 represents "Neutral", 5–7 mirrors the ordinal pattern for positive responses. Our descriptive

analysis found that the distribution of ordinal responses was heavily left skewed, where the proportion of the top two scores (i.e. 6 –"Enjoyed" and 7 –"Enjoyed very much") accounting for roughly half of overall responses (see distribution histogram in *S3 Appendix* and descriptive statistics of the distribution in *S4 Appendix*). As such, the measure was transformed into a binary variable with enjoyment six and above defined as high ExWB, five and below defined as average-to-low ExWB. The use of binary ExWB measurement seems novel in literature and has two advantages. Firstly, the ordinal enjoyment measurement is difficult to compare across individuals and years, given its highly context-dependent nature. Using a binary variable could, to some extent, reduce the volatility and increase the clinical relevance of enjoyment measurement between survey periods. It enabled us to focus on identifying determinants that may lead to higher-than-average ExWB, as opposed to factors that may lead to a general increase of enjoyment. Secondly, it reduces the computational complexity for model estimation.

## Statistical analysis

Latent class analysis (LCA) is used to identify nuanced lifestyles within each working mode group (homeworkers, commuters, and hybrid workers). LCA is a generalised structural equation model used to identify a latent (unobserved) categorical variable based on a set of observed categorical variables (Sinha et al., 2021). It is used to find groups with similar features or to decipher distinct patterns in multivariate data through maximizing cross-group variance and minimising within-group variance. The goal is to simplify and visualise complex and high-dimensional data in an unsupervised manner. Identification of LCA classes is particularly useful in large-scale observational studies and helps contextualise unobserved heterogeneity. In particular, the relatively homogenous groups identified as a result of the LCA models have distinctive spatial and temporal features representing interpretable lifestyles. The identification of lifestyle delineates heterogenous subgroups of the sample that would have otherwise aggregated as a homogenous sample, mitigating the risk of inadvertently averaging out important heterogeneities in these unidentified latent subgroups. For key formulae and technical specifications see *S5 Appendix*.

We estimated latent class models using robust maximum likelihood estimation with Newton-Raphson stepping mechanism, repeated for one to six latent classes. The 24 hourly categorical variables indicating activity type were used as explanatory variables in the models, together with socioeconomic covariates including income quantile, education attainment, and occupation classification. To help with model convergence, we used a two-step integration process: Laplacian approximation was used in the first round of integration, the results matrix was then used as starting value for the more accurate mean-variance adaptive Gauss-Hermite quadrature integration method. The final class size within each working mode was determined using the Akaike's information criterion and Bayesian information criterion values (see model fit estimations in *S6 Appendix*), average latent class probabilities, and substantive interpretation of the classes identified. The LCA was conducted using pooled samples from both pre- and during COVID-19. After LCA, the samples were then separated into pre- and during COVID groups according to the survey date.

Logistical regression models are used to explore potential factors that contribute to higher-than-average ExWB. The dependent variable is the binary ExWB (1 for higher ExWB and 0 for lower and average ExWB), and the independent variables in the model are broad activity type, lifestyle, the interaction term between broad activity type and lifestyle, socioeconomic covariates including sex, age group, income group, education attainment, occupation classification

and residential region, and month of survey collection to control for idiosyncratic differences. Results for pre- and during COVID data are compared.

## Conceptualising spatio-temporal flexibility

A two-dimensional conceptual framework for measuring working flexibility is proposed, aiming to explicate the heterogenous wellbeing changes observed across the different lifestyles. The spatial dimension measures the worker's ability to choose the location to engage in paid work on an intraday basis. Spatial flexibility is a normalised entropy (see key formulae specified in *S7 Appendix*) measure of where workers are located during the daytime activities window. The key variable measured is the location of an individual between the first and last hour for paid work. All locations recorded within the period are included in the entropy calculation, regardless of activity type.

Temporal flexibility is a distinct but interrelated dimension of flexibility, which concerns the ability to choose when to conduct paid work on an intraday basis, which is linked to how workers organise their non-work activities including travel. The temporal flexibility is quantified using a normalised entropy (see key formulae specified in *S7 Appendix*) measure of when breaks related to Personal and Leisure activities take place during the working hours, e.g. lunch breaks and short strolls. The daily working hours is identified by the timestamp of the first and last episode of paid work. One will achieve high temporal flexibility if they engage in numerous or prolonged breaks during the working hours. A higher entropy would derive from numerous or prolonged enjoyable breaks, indicating greater ability to organise one's time, hence greater temporal flexibility.

## Results

A total of ten lifestyles were identified across the three working modes: three within homeworkers, four within commuters and three within hybrid workers. Following the tempogram method developed by Kolpashnikova et al. [35], Fig 1 shows the distinct intraday time use patterns of different lifestyles and the corresponding intraday ExWB patterns. Table 1 further provides the pen portrait for each lifestyle. The naming of *lifestyles* is primarily based on paid work schedules because of its dominant role in lifestyle choice and identity in Anttila et al. [10]. Sociodemographic characteristics of each lifestyle are summarised in *S8 Appendix*.

The defining time use pattern for each lifestyle remained largely consistent pre- and during COVID-19, though the sample distribution across lifestyles changed significantly, notably a significant shift from Conventional Commuter (CC, N = 1,046 pre-COVID) to Emergent Homeworker (EH, N = 482 during COVID). The direction and magnitude change in ExWB during the COVID-19 pandemic were heterogenous across time of the day, activity types and lifestyles. We discuss these aspects in turn.

To better understand the heterogeneity observed across the time of the day, we bifurcate the diary day into two 12-hour windows: for daytime activities (between 6am and 6pm) and for evening activities (before 6am and after 6pm). The division recognises the stratifying influence of institutional-organisational time [36]. As we use full employment as an inclusion criterium, within-sample must account for collective time schedules–such as those commonly prescribed by the employer during the daytime activities [37]. We focus primarily on the daytime activities window (between 6am and 6pm) as significant heterogeneity is observed across activities and lifestyles during these hours. In addition to the empirical observation that informed the selection of temporal boundaries for daytime and evening, this window maximises the diversity and richness of daytime activities captured by the granular intra-day temporal data. It expands on the conventional paid work window (such as 9am-5pm), which

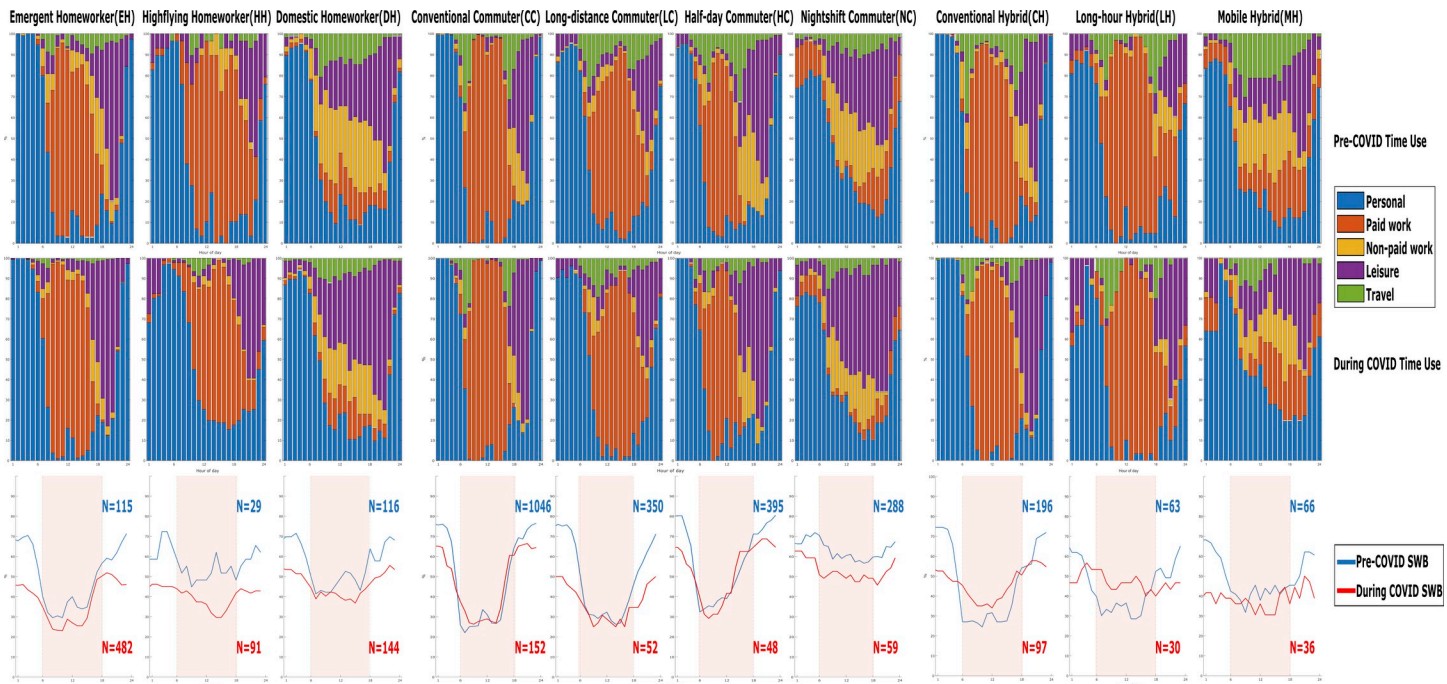

**Fig 1.** Intraday patterns of time use (top two rows) and ExWB (bottom row)*. * The x-axis represents 24 hours of the day for all charts. For time use patterns, the y-axis is the percentage proportion of the samples engaged in each of the five time use categories summing to 100%. For ExWB patterns, the y-axis is the percentage proportion of the samples reporting high ExWB. The daytime activities window are highlighted for ExWB charts.

excludes important activities such as travel and non-paid work which reflect the distinct life-styles we identified. The 6am and 6pm temporal boundaries result in two comparable observational windows which refine our focus to improve understanding of ExWB and flexible working.

## Time use and ExWB during the evening activities window

During COVID, ExWB declined for all lifestyles within the evening activities window when most people engaged in leisure or personal activities such as sleeping. However, time use pattern during these hours remained virtually unchanged pre- and during COVID, which suggests that the ExWB decline during these hours seemed uncorrelated with time use and may be attributed to general anxiety and potentially blurred boundary between work and life associated with flexible working. This finding emphasises the need to investigate nuanced intraday variation of subjective wellbeing through separating the daytime activities from the evening activities. It could be postulated that the uniform decline of ExWB within the evening activities window was a main contributor to the overall deterioration of mental health in the UK during COVID-19. A rapid and significant rebound of ExWB after paid work was observed consistently across lifestyles before the pandemic. During COVID-19, however, the rate of ExWB rebound in the early evening flattened, particularly for homeworkers and hybrid workers who tend to be more susceptible to the effect of blurred boundary between work and life.

## Time use and ExWB during the daytime activities window

For most lifestyles, paid work is concentrated during the daytime activities window. Furthermore, negative ExWB changes during paid work were observed in most lifestyles. Enjoyment tended to plummet at the start of daytime activities window, with slight uptick in mid-day

**Table 1. Lifestyle pen portraits.**

| | Working Mode | Homeworker | Commuting Worker | Hybrid Worker |
|---|---|---|---|---|
| **Lifestyle** | | Paid work engaged exclusively in the home environment. Younger and single samples increased during COVID-19, signalling democratisation of homeworking. | Paid work engaged exclusively in the workplace environment. Demography became older and generally worse off economically during COVID-19, most of whom are essential workers. | Paid work split between home and workplace. The proportion of respondents in hybrid work mode remained consistent pre- and during COVID-19. |
| **Early Concentrated Working** | Most start work around 7-8am and finish before 6pm. Their paid work hours are highly concentrated around the conventional working hours. | *Emergent Homeworker (EH)* Becomes the dominant lifestyle during COVID-19 representing 40% of all respondents (70% of homeworkers), a tenfold increase from its pre-pandemic levels. | *Conventional Commuter (CC)* Dominant lifestyle pre-pandemic representing 39% of all respondents (50% of commuters), during COVID-19 it is the second most populous lifestyle (13% of all & 46% of commuters). | *Conventional Hybrid (CH)* Highly male dominated both pre- and during COVID-19, it is the dominant lifestyle within the hybrid work mode (8% of all & 60% of hybrid). |
| **Long Dispersed Working** | 9-10am sees the majority starting paid work, though a large number work late until 8pm. Their working hours are more disperse and flexible. They spend the least time on non-paid work both before and during COVID-19. | *Highflying Homeworker (HH)* Consistently has the highest share of high-income earners. They spend little time on commuting, and more time on paid work and personal maintenance. | *Long-distance Commuter (LC)* Spends the longest duration on commuting, and very little time on non-paid work. They have leisure activities late into the evening and tend to be younger. | *Long-hour Hybrid (LH)* The only lifestyle to spend more than 10 hours in paid work both pre- and during COVID-19; spends consistently little time on sleep and other personal maintenance activities. |
| **Non-work Dominated** | Non-paid work and leisure dominate daily schedule, while engaging in a fraction of paid work hours as other lifestyles. Travel is flexible and not concentrated around the peak commute windows. | *Domestic Homeworker (DH)* Spends the least amount of time in paid work and most time in non-paid work consistently pre- and during COVID-19. One of the few lifestyles dominated by women. | | *Mobile Hybrid (MH)* Spends significant amount of time on commute and non-paid work; highest share of middle-aged respondents pre-COVID-19; but the share of married young respondents increased significantly during COVID-19. |
| **Half-day Working** | Only identified within commuters. Paid work starts very early before 7 am and ends around noon. Their afternoons have high concentrations of leisure. | | *Half-day Commuter (HC)* High proportion of manual workers, and low proportion of managers; has the lowest proportion of undergrad degree holders. Demography was heavily skewed toward older age groups. | |
| **Nightshift** | Only identified within commuters. Paid work activities are concentrated in the evening hours, their mornings are dominated with non-paid work and long leisure durations in the afternoons. | | *Nightshift Commuter (NC)* Greatest decrease of high earners during COVID-19, as a result, it has the highest portion of low earners and only 2% in the high earner category. | |

which coincided with usual lunch breaks, and some recovery after work. The uptick during lunch break was notably less notable during COVID-19 and even disappeared for some lifestyles (e.g. HH, DH and CH). Lifestyles with more concentrated paid work periods such as EH, CC, and CH would experience greater intraday ExWB fluctuations. To quantify the varying changes while controlling for sociodemographic covariates, logistical regression is applied to activity episodes within the daytime activities window. Results are reported in Table 2, for full model results see *S9 Appendix*.

Before the COVID-19 pandemic, homeworkers, particularly the highflying homeworkers (HH), tended to have better ExWB than conventional commuters (CC, as the reference case). ExWB of hybrid workers was not statistically higher than CC. It implies that homeworking may be a privilege for some (e.g. HH), but hybrid working appeared a compromise for those who had to juggle paid work and domestic duties. However, during COVID-19, relative ExWB (based on CC) of all homeworking lifestyles decreased significantly, particularly the EH

**Table 2. Logistical regression models (within the daytime activities window & workdays only).**

| | | Pre-COVID-19 | | During COVID-19 | |
|---|---|---|---|---|---|
| | | Coefficient | Robust standard error | Coefficient | Robust standard error |
| *Broad Activity Type* | Personal | **0.54***** | 0.04 | **0.98***** | 0.06 |
| | Paid Work | **Ref** | - | **Ref** | - |
| | Non-paid Work | **0.34***** | 0.05 | **1.03***** | 0.10 |
| | Leisure | **1.23***** | 0.07 | **1.34***** | 0.09 |
| | Transport | **-0.05** | 0.05 | **0.55***** | 0.07 |
| *Lifestyle* | EH | **0.21**** | 0.09 | **-0.15***** | 0.03 |
| | HH | **0.84***** | 0.14 | **0.12***** | 0.05 |
| | DH | **0.26**** | 0.13 | **-0.12** | 0.07 |
| | CC | **Ref** | - | **Ref** | - |
| | LC | **-0.24***** | 0.06 | **-0.17***** | 0.06 |
| | HC | **0.33***** | 0.06 | **0.19***** | 0.06 |
| | NC | **0.22*** | 0.13 | **-0.37**** | 0.18 |
| | CH | **-0.09** | 0.06 | **0.34***** | 0.04 |
| | LH | **0.16*** | 0.10 | **0.39***** | 0.06 |
| | MH | **0.15** | 0.17 | **-0.39***** | 0.12 |

group, the majority of whom were likely new homeworkers transitioned from commuters as a result of lockdown measures. By contrast, relative ExWB increased for all hybrid lifestyles except the Mobile Hybrid (MH), who experienced a significant decline of ExWB relative to CC. MH lifestyle features the shortest paid work and longest non-paid work both pre- and during COVID-19 (see *S8 Appendix*). The decrease of ExWB for homeworkers and increase of ExWB for hybrid workers combined suggest that the optionality over place of work and the ability to continue to travel during the pandemic seemed to bring ExWB betterment. However, for hybrid workers who bear considerable domestic duties, the positive ExWB effect associated with such optionality would diminish and may even turn into negative if fulfilling certain domestic duties (e.g. home schooling, caring duties outside own home) became increasingly difficult during the pandemic.

Other plausible factors that may contribute to the ExWB increase of hybrid workers during the pandemic include social interaction opportunities at the workplace, the ability to stay outdoor as part of travel, and the formalisation of hybrid working as an acceptable practice. It is also found that the ExWB increase for hybrid workers stem from enjoyment boost associated with some activity types but not all, which will be investigated in the next section.

### Heterogeneity of ExWB changes across activity types and latent lifestyles

Fig 2 shows average ExWB changes from pre- to during COVID-19 within the daytime activities window by activity type and lifestyle. In terms of the direction of ExWB, it decreased during paid-work for almost all lifestyles, with a notable exception of the long-hour hybrid lifestyle (LH), which might be attributed to the increasing appreciation of certain occupations during the pandemic. ExWB decline on leisure activities was observed for almost all lifestyles, despite an increase of time spent for leisure for all lifestyles (see *S8 Appendix*). Lee & Tipoe [9] pointed out that the lack of company and social interactions in leisure activities may be a main cause of ExWB decline.

By contrast, ExWB associated with travel activities has increased for most lifestyle groups except the Domestic Homeworker (DH), who spent more time on transport than other homeworkers and on par with some commuters (see *S8 Appendix*). For DH, the need to engage in

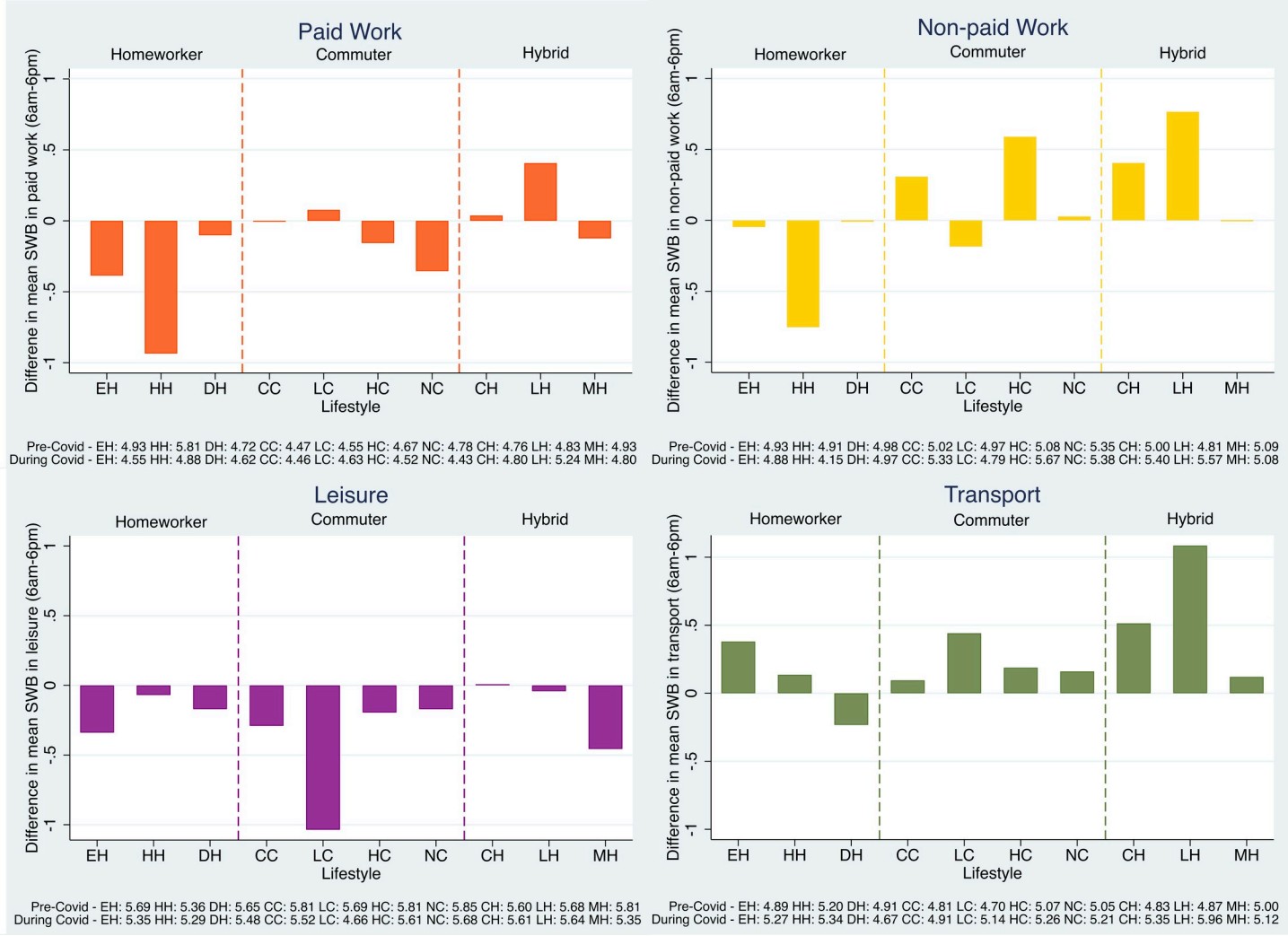

**Fig 2. Difference in mean ExWB within the daytime activities window by activity type.** * The y-axis represents the difference in average ExWB for each lifestyle while engaged in the respective activity type. Pre- and during COVID-19 average values used to calculate the differences are in the legend.

essential trips (e.g. to pharmacies and caring duties) may cause stress and anxiety due to difficulties in travel and the risk of contagion. ExWB increase from moderate travel activities, particularly for hybrid workers (CH & LH) during the pandemic corroborates the early finding that the optionality over place of work and the ability to continue to travel had a positive ExWB effect. For non-paid work, ExWB increased for some commuter and hybrid worker lifestyles, but decreased for all homeworker groups, implying emerging difficulties in (re-)balancing life and work for homeworkers.

During COVID-19, five out of six large ExWB changes (> = 0.5 or < = -0.5) occurred in late and dispersed lifestyles: HH, LC, and LH. This is contrasted by the relatively low magnitude of changes for the non-work dominated lifestyles: DH, HC, NC, and MH. These lifestyles tended to be less constrained by established work-time regime, enabling a higher degree of flexibility for adjusting their time use to mitigate negative ExWB changes observed during the pandemic.

## Conceptualising working flexibility–A time use perspective

Our analysis shows COVID-19-related changes in ExWB is heterogeneous in terms of the direction and magnitude for different activity types and lifestyles. The identification of latent lifestyles and the comparison of ExWB changes across lifestyles shed a new light on the conceptualisation of 'flexibility' from a time use perspective. Specifically, flexibility could be conceptualised over two dimensions, spatial flexibility and temporal flexibility [10]. Based on two normalised entropy measures respectively for spatial and temporal flexibilities, we visualise a range of spatio-temporal flexibilities for each lifestyle in Fig 3.

Spatial flexibility is closely associated with working modes as it is a normalised entropy measure of workplace diversity. Homeworkers (hybrid workers) tend to have the lowest (highest) spatial flexibility as depicted along the horizontal axis in Fig 3. Temporal flexibility is a normalised entropy measure of when potentially enjoyable breaks take place over the working day. Potentially enjoyable breaks are defined as Personal or Leisure activities during the working day (e.g. lunch break or a short stroll). Temporal flexibility is closely associated with lifestyle–all three long and dispersed working lifestyles (HH, LC, LH) and two non-work dominated lifestyles (DH & MH) have consistently high temporal flexibility pre- and during COVID-19.

The spatio-temporal framework provides a novel perspective to understand the relationship between wellbeing and flexible working. Existing discourse on flexible working has been predominantly focused on the spatial dimension (e.g. in the form of hybrid working), with an underlying assumption that the higher flexibility the better for workers. However, our analysis suggests that hyper spatial flexibility, combined with hyper temporal flexibility, may be detrimental to ExWB. For example, the Mobile Hybrid (MH) lifestyle featured high flexibility in both dimensions, but their ExWB was not statistically higher than Conventional Commuters (CC) pre-pandemic (see Table 2). During the pandemic, the MH lifestyle (mostly married,

**Fig 3. Spatio-temporal flexibility by working arrangement & lifestyle (adopted from Wan & Chen [38]).** *The axes are normalised entropy measures for spatio-temporal flexibilities. Each lifestyle has two data points, one calculated using data from pre-COVID-19 data and the other during COVID-19. The two points are connected to represent the range of spatio-temporal flexibility between pre-pandemic normalcy and changes induced by COVID-19.

young respondents–see *S8 Appendix*) experienced a notable decrease in ExWB relative to CC. By contrast, lifestyles of relatively high temporal flexibility and low-to-moderate spatial flexibility showed positive ExWB effect pre-COVID-19.

The implication is twofold: firstly, the marginal utility of greater spatio-temporal flexibility diminishes and may even turn negative (e.g. working across many locations to the point that breaks are no longer enjoyable but a mere necessity); secondly, the hyper spatial-temporal flexibility, as captured by the proposed measurements, may represent a compromise, rather than a voluntary lifestyle choice. The negative wellbeing change associated with such involuntariness exacerbated during the pandemic, potentially due to imposed furlough and/or increasing domestic duties such as home-schooling.

To postulate the evolving relationship between ExWB and flexible working in a post-pandemic context, spatial flexibility is likely to increase for all lifestyles due to recovered confidence for travel, while the change of temporal flexibility tends to be more complex across lifestyles. On the one hand, relatively high temporal flexibility as an outcome of voluntary lifestyle choice may explain the positive wellbeing changes, hence the desirability of homeworking and hybrid working, but such positive changes may be rather limited if a more temporally flexible working pattern was not offered by employers. On the other hand, the non-linear relationship between spatial-temporal flexibility and ExWB suggests that workers may benefit from the optionality of multiple working modes across days of the week, e.g. reduced (increased) working hours for some (other) days of the week. The lifestyle perspective based on time-use patterns seems effective to capture such nuances.

## Limitations

The study samples only include those in full-time employment, as our aim is to identify latent lifestyles amongst a comparable pool of full-time (self-reported) workers. Including part-time workers would potentially confound the identification and interpretation of the latent classes. We identified groups of workers who are in full-time employment with distinctly heterogenous spatial and temporal working patterns. Had self-declared part-time employees been included in the LCA models, we would not be able to separate these nuanced lifestyles amongst full-time workers. Self-employed workers are also excluded from the LCA models, as their flexibility to greater extents are self-determined. This is particularly relevant for the policy implications, as we argue against blanket "back-to-workplace" policies–these are most applicable to those employed on a full-time basis rather than the self-employed workers.

A minority of lifestyles identified from LCA models have the limitation of small class sizes, in particular two of the hybrid lifestyles during the pandemic (LH and MH have 30 and 36 individuals respectively). Compared to lifestyles with more individuals, the ExWB averages produced may be more prone to the effects of outliers. While the logistical regression produced highly statistically significant estimates for the smaller LH and MH lifestyles during the pandemic (>99% significance), it does not rule out possible limitation affecting the external validity of these estimates due to smaller sample sizes. This limitation resulted from a confluence of a minority of workers engaging in flexible working lifestyles such as hybrid lifestyles, as well as the data restriction of fewer individuals in the during COVID dataset. As hybrid working becomes more prevalent and more post-pandemic data is gathered, it would be meaningful to verify the external validity of the findings for the smaller LCA classes of hybrid workers.

The logistical regression model does not include parental status and household composition as covariates. The primary consideration for excluding household composition (including parental status) is a significant non-response issue in the original data where 20% of study samples did not provide the number of children. Inclusion of the covariate would significantly

reduce the samples size, leading to identification issues. However, this exclusion may give rise to endogeneity in the logistical regression model. Firstly, childcare and eldercare hours are highly endogenous to the enjoyment of non-paid work, comparing the enjoyment levels during non-paid work activities without explicitly controlling for care duty may risk confounding the coefficients estimated. Secondly, household composition and parental status are important sociodemographic characteristics, the exclusion may lead to incompleteness in the model parameters. Thirdly, without controlling for differences in household composition and parental status weakens the generalisability when comparing ExWB changes before and after the pandemic. Further research is needed to expand the observational unit to the household rather than at the individual level to address the complexities and better address potential endogeneity.

The lack of longitudinal data limits our ability to control for employment change across waves, which we addressed by restricting samples to full-time workers at each data cross-section for comparability. On an aggregate level, the logistical regression model results cannot capture changes in the labour market and the associated background effects on ExWB. Specifically, the background effects may be linked to an upward bias in ExWB at the start of the pandemic, as our samples were fully employed despite the labour market contraction. If such background effects were present and uniformly impacted samples during COVID-19, the relative differences in ExWB across lifestyles would remain constant. On the other hand, non-uniform background effects may have contributed to, and help explain, the observed heterogeneity between lifestyles. In either case, background effects associated with labour market changes would not contradict our core findings. Longitudinal data would be required to identify the presence of such effects, and to delineate the sources of heterogeneity.

Furthermore, due to the non-interventional data, our study is not able to establish causal relationships–all findings are hence associative. Important issues such as the direction of causation and potentially circular causality cannot be explicitly modelled without longitudinal (and interventional) data. Using cross-sectional data, the possibility of self-selection bias cannot be ruled out. It is possible that workers with higher (lower) well-being are more inclined to work from home (office), and the ExWB difference arises not from the lifestyle choice of particular spatial or temporal working arrangements. However, the two-dimensional flexibility argument posits that if workers are able to self-select where and when they work, it could be a mechanism to avoid detriments to ExWB. Associative findings from our studies offer important insights and a promising avenue for future causal investigations.

Another significant limitation of the study is the single measurement of ExWB. A more comprehensive measure of subjective wellbeing would be ideal. This could potentially be achieved through implementing a supplemental questionnaire on long-term mental wellbeing following the original time-use survey. Standardised TUS data is collected in more than 30 countries. If more countries incorporate an ordinal ExWB variable and an additional questionnaire on long-term mental wellbeing, a large-scale international comparison could be conducted, which may shed light on the changing future of work from a public health perspective. Furthermore, the data from the UKTUS are population-representative but nevertheless cross-sectional. Higher quality longitudinal data, such as a cohort study would enable modelling analysis of causal relationship between lifestyles and ExWB.

## Conclusion

This paper aims to investigate nuanced and evolving wellbeing changes associated with COVID-19 across distinct lifestyle groups of workers in the UK. The main strength of the study is our novel lifestyle perspective enabled by the time-use survey data collected before and

during COVID-19. Through identifying latent but distinct lifestyles of workers in the UK, our research provides timely and fresh insights on the relationship between wellbeing and flexible working and the heterogenous changes associated with COVID-19 on across activity types and lifestyles. We found the direction and magnitude of ExWB changes were not uniform across activity types, time of day, and lifestyles. ExWB changes during the evening activities window were consistently negative for all lifestyles. In contrast, the direction of wellbeing change during the daytime activities window varied across lifestyles. It is also demonstrated that a spatio-temporal conceptualisation of working flexibility may help explicate the strong and non-linear correlations between ExWB and lifestyles. While flexible working (e.g. in the form of 4-day work week) is gaining and retaining popularity for some sectors and occupations [39], our empirical finding contests the simplistic assumption that the higher working flexibility the better. Homeworking is not inherently more flexible, especially when temporally inflexible work-time regime persists. The non-linear relationship identified emphasises the imperative of developing a proper conceptualisation of working flexibility, which would be crucial for informing the 'back-to-work' policies.

One key policy implication is that existing discussion on 'back-to-work' policies should expand beyond the spatial dimension of working flexibility (i.e. where to work) and focus more on the temporal dimension in relation to more nuanced segmentations of the workforce. Interventions to increase temporal working flexibility may lead to higher worker wellbeing and resilience to tackle negative shocks in the labour market. A lifestyle perspective based on time-use patterns seems provide a viable approach to link the spatial-temporal flexibility with subjective wellbeing and potentially other socio-economic outcomes, notably labour productivity. 'Back-to-work' policies should provide wider support for lifestyle adaptation and transitions.

## Supporting information

**S1 Appendix. Sample distribution amongst survey waves.**
(DOCX)

**S2 Appendix. Aggregating activities into time use categories.**
(DOCX)

**S3 Appendix. ExWB distribution histogram.**
(TIFF)

**S4 Appendix. ExWB descriptive statistics.**
(DOCX)

**S5 Appendix. Key formulae for LCA.**
(DOCX)

**S6 Appendix. LCA model fit–information criterion.**
(DOCX)

**S7 Appendix. Key formulae for normalised entropy measure for spatio-temporal flexibility.**
(DOCX)

**S8 Appendix. Sample descriptive statistics.**
(DOCX)

**S9 Appendix. Full logistical regression model results.**
(DOCX)

## Author Contributions

**Conceptualization:** Jerry Chen, Li Wan.

**Data curation:** Jerry Chen.

**Formal analysis:** Jerry Chen.

**Investigation:** Jerry Chen, Li Wan.

**Methodology:** Jerry Chen, Li Wan.

**Resources:** Li Wan.

**Software:** Jerry Chen.

**Supervision:** Li Wan.

**Validation:** Jerry Chen.

**Visualization:** Jerry Chen.

**Writing – original draft:** Jerry Chen.

**Writing – review & editing:** Li Wan.

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
