## [Decision Letter · Decision Letter 0]

19 Jun 2023

PONE-D-23-04776Remote working and experiential wellbeing: A latent lifestyle perspective using UK Time Use Survey before and during COVID-19PLOS ONE

Dear Dr. Wan,

Thank you for submitting your manuscript to PLOS ONE. After careful consideration, we feel that it has merit but does not fully meet PLOS ONE’s publication criteria as it currently stands. Therefore, we invite you to submit a revised version of the manuscript that addresses the points raised during the review process. I am particularly concern with methods, in addition that the paper needs clarity and the inclusion of recent  economic literature on remote work. Authors use two cross-sections of UKTUS and then they can derive corretations. Consequently, the redaction shoud be appropriate in this context (e.g. avoiding causal terms as "impact").

We look forward to receiving your revised manuscript.

Kind regards,

José Alberto Molina

Academic Editor

PLOS ONE

Journal Requirements:

Reviewers' comments:

Reviewer's Responses to Questions

**Comments to the Author**

1. Is the manuscript technically sound, and do the data support the conclusions?

Reviewer #1: Partly

Reviewer #2: Yes

2. Has the statistical analysis been performed appropriately and rigorously? 

Reviewer #1: No

Reviewer #2: Yes

3. Have the authors made all data underlying the findings in their manuscript fully available?

Reviewer #1: Yes

Reviewer #2: Yes

4. Is the manuscript presented in an intelligible fashion and written in standard English?

Reviewer #1: Yes

Reviewer #2: Yes

5. Review Comments to the Author

Reviewer #1: Referee’s Report

Manuscript PONE-D-23-04776

“Remote working and experiential wellbeing: A latent lifestyle perspective using UK Time Use Survey before and during COVID-19”

This paper studies the relationship between experiential well-being (ExWB) and remote work, while controlling for other relevant factors, such as hours of work, occupation, and family-related variables, such as marital status. Using time surveys from the UK, before and during the COVID-19 pandemic, the authors employ latent class models to examine the heterogeneous impacts of ExWB across distinct activity types, time of day, and lifestyles.

Overall, the paper is well-written and presents interesting findings. However, there are several concerns regarding the study's contribution and the validity of the results. These concerns may require further clarification and elaboration in order to strengthen the study’s contribution and robustness.

Concerns

1. The abstract is quite long. The authors should focus on the most important finding(s) of their paper.

2. Authors examine the relationship between experiential well-being (ExWB) and remote work, including in their analysis other factors, such as employment characteristics, i.e., hours of work, occupation, or family factors, such as marital status. There is literature that relates parental status (motherhood/parenthood) to remote work and well-being. Authors should justify why they haven’t included in their analysis any indicators for parental status, for example number of children.

3. “The direction and magnitude of pandemic impact on ExWB were heterogenous across time of the day, activity types and lifestyles”. Authors acknowledge the heterogeneous effects of socio-economic factors on ExWB; they should explain better how latent class models help to deal with these heterogeneous effects in the present study.

4. I have concerns about how novel is the use of latent class models and time-use survey data. For example:

Lee, Y. Activity Profiles among Older Adults: Latent Class Analysis Using the Korean Time Use Survey. Int. J. Environ. Res. Public Health 2021, 18, 8786. https://doi.org/10.3390/ijerph18168786, uses latent class models along with time-use survey data in a very similar way. Authors should acknowledge relevant to the use of latent class models literature and time-use survey data and motivate better their own contribution.

5. Looking at the number of total observations for each class (Appendix 5), I feel that the authors end up with too many classes and in some cases too few observations for each class. This study would probably benefit from other measures of model of fit and fewer classes or even a different method. These combined can provide to the analysis higher statistical power and more robust results.

6. The authors claim that ExWB impact outside of usual working hours (before 6am and after 6pm) was consistently negative for all lifestyles. The choice of these thresholds seems quite arbitrary. Have other authors used such thresholds in the past? Why not for example before 9am and after 5pm? The authors should cite appropriately other authors that have used these thresholds or/and explain better and more the rationale of their choice.

7. I have concerns about the direction of causality between flexible/remote working and well-being. Some of the data were collected during the pandemic when mental health pressures were exceptionally high. Specifically, I am wondering whether flexible/remote working leads to changes in well-being, or if individuals with better (or worse) well-being are more inclined to work less (or more) flexibly/remotely and travel less (or more). Given the potential for reverse causality, it would be beneficial for this study to include a discussion on the direction of causality between flexible/remote working and ExWB.

While the authors do acknowledge some of the limitations of their study, they do not address potential issues of reverse causality or/and self-selection. So, it is strongly recommended that the authors explore alternative explanations for the observed associations, such as the potential for individuals with better (or worse) well-being to self-select into more (or less) flexible working arrangements.

8. “In this paper, the ExWB measure was transformed into a binary variable with enjoyment six and above defined as high ExWB, five and below defined as average-to-low ExWB.” The authors should explain the rationale of this choice. Have other authors used the same/similar definition? The authors should provide the specific wording/questions used from the time-survey(s) to construct their ExWB measure. Have authors tried alternative definitions? It would make sense for example to have a binary variable equal to 1 for enjoyment levels (5-7), and 0 otherwise, especially when the mean is a little bit above 5.

9. BAME stands for Black, Asian, and Minority Ethnic groups, not Black, Asian, Middle Eastern groups as stated. Also, for robustness authors can think BME groups (Black and Minority Ethnic) only.

10. It is not clear why authors have used only UKTUS 2015, and the first online wave collected in 2016 to capture the pre-pandemic time-use. The authors should provide more information about any non-availability of data. For example, is this non-availability because the survey is conducted every 5 years? Aren’t data for other months from 2020 or 2021 available? In the case that authors have purposefully selected to use data for specific months they should justify adequately this selection.

11. What is full employment? Full-time employment? If this is the case, is there any reason why part-time workers were excluded from the analysis? If specific groups of workers were excluded, then the possibility of sample selection bias arises. Women, for example, tend to work part-time to a greater extent than men, which could potentially pose a challenge to the study's validity. What about self-employed? Are they considered as home workers? Have self-employed been included or excluded?

12. “The longitudinal dimension is enabled by combining population-representative, repeated cross-sectional time use data gathered during the pandemic (2020) with the pre-pandemic UK Time Use Survey (UKTUS) in 2015. I am not clear about the longitudinal dimension of the data. My impression has been that the data is cross-sectional. Authors should provide more details about the longitudinal dimension of the data, if there is any such a dimension.

13. According to Appendix 3, the distribution is left-skewed (skewness is negative); not right skewed as stated.

14. To motivate better their contribution and the importance of their findings, authors should think to provide literature and more evidence of how spatial and temporal flexibility looked like before and during the pandemic.

15. In order to enhance the readability of the results, it is recommended to include the initials for lifestyles, which are reported in Table 2, in parentheses in Appendix 5. This would facilitate the comprehension of the findings presented in the Appendix.

Reviewer #2: Remote working and experiential wellbeing: A latent lifestyle perspective using UK Time Use Survey before and during COVID-19

This paper looks at the heterogeneous effects of wellbeing effects of the COVID-19 pandemic amongst the workforce in the UK. It specifically defines heterogeneous effects across lifestyles and work modes instead of the usual characteristics that have been used for this purpose in the literature (i.e. gender, ethnicity, income etc). Using the UKTUS the study found that experiential wellbeing is heterogeneous across workers lifestyle.

I agree with the authors on the importance of the study and the uniqueness of the time use data they have used. I generally think the paper is well-written and makes an important contribution by showing that both spatial and temporal flexibility is related to wellbeing.

I think the authors can improve on the clarity of the paper, I don’t think it will be easy for another author to replicate their result based on the description. For example, the implementation of the normalized entropy is not clear to me at all from the description. What entropy formulae was used and how exactly was it implemented? Similarly, since Plos one is interdisciplinary it might be useful to explain a bit more how the latent class analysis works and what it seeks to achieve. Lastly, the statistical analysis section mentioned that the LCA was conducted using pooled samples from pre-and during COVID. Why was this done this way? Why not have separate analyses for the two periods given that the dynamics can be different? To what extent does this affect the conclusions i.e. is the result robust to the choice of using pooled sample?

Further, the description of Table 2 (for logistical regression) suggests that there are additional controls including the interaction between activity type and lifestyle. Mentioning the interaction term for example will create the impression that these interaction coefficients are important (especially if they were significant) but the table neither shows this result nor discuss them. A related point is that when a model includes interaction terms, I don’t think the main effects can be interpreted the same way one would have without interaction terms, which then makes not presenting the full model’s result problematic.

I think if these concerns can be addressed the paper makes an important contribution to the literature.

6. PLOS authors have the option to publish the peer review history of their article (what does this mean?). If published, this will include your full peer review and any attached files.

Reviewer #1: No

Reviewer #2: **Yes: **Adeola Oyenubi (PhD)

---

## [Author Response · Author response to Decision Letter 0]

23 Aug 2023

Responses to Reviewers

We would like to thank the Reviewer for taking time to review our manuscript and for the constructive comments provided, which help us improve the quality of the paper. Please find below our point-to-point responses.

Responses to the Editor

1. I am particularly concern with methods, in addition that the paper needs clarity and the inclusion of recent economic literature on remote work. Authors use two cross-sections of UKTUS and then they can derive correlations. Consequently, the redaction should be appropriate in this context (e.g. avoiding causal terms as "impact").

Response: Thank you for the cogent comments. We confirm that the study is focused primarily on correlations. We have thus refined our wording to avoid alluding to causality in the manuscript. We also added the below paragraph on recent economic literature related to remote work:

“A systematic review on the impact of remote work on productivity found stark contrast where before the pandemic 79% studies showed positive effects of remote work, that figure dropped to 23% during the pandemic (Hackney et al., 2022). Preliminary evidence show that remote workers experience more anxiety around job security (Fosslien & Gottlieb-Cohen, 2023), and are more prone to loneliness (Miyake et al., 2022). Cell-phone-based mobility data reveal trips to the CBD stabilising at 60% of pre-pandemic levels, suggesting remote workers will make up a significant portion of the future workforce (Monte et al., 2023).”

Response: Our data is openly available through the UK Data Service. The DOIs are included under the “Data repository” section. The STATA codes are available via our GitHub link, also included under the “Code availability” section of the manuscript.

Response: Thank you for outlining the additional requirements. The supporting information files and the in-text citations have been adjusted to the journal guidelines.

 

Responses to Reviewer #1 

This paper studies the relationship between experiential well-being (ExWB) and remote work, while controlling for other relevant factors, such as hours of work, occupation, and family-related variables, such as marital status. Using time surveys from the UK, before and during the COVID-19 pandemic, the authors employ latent class models to examine the heterogeneous impacts of ExWB across distinct activity types, time of day, and lifestyles.

Overall, the paper is well-written and presents interesting findings. However, there are several concerns regarding the study's contribution and the validity of the results. These concerns may require further clarification and elaboration in order to strengthen the study’s contribution and robustness.

Concerns

1. The abstract is quite long. The authors should focus on the most important finding(s) of their paper.

Response: Thank you for this cogent suggestion. The abstract has been reduced from 279 words to 207 words. The updated abstract is quoted here for your reference:

“Mental health in the UK had deteriorated compared with pre-pandemic trends. Existing studies on heterogenous wellbeing impacts of COVID-19 tend to segment population based on isolated socio-economic and demographic indicators, notably gender, income and ethnicity, while a more holistic and contextual understanding of such heterogeneity among the workforce seems lacking. This study addresses this gap by 1) combining UK time use surveys collected before and during COVID-19, 2) identifying latent lifestyles within three working mode groups (commuter, homeworker and hybrid worker) using latent class model, and 3) quantifying nuanced experiential wellbeing (ExWB) impacts across workers of distinct lifestyles. It was found that the direction and magnitude of ExWB impact were not uniform across activity types, time of day, and lifestyles. The direction of ExWB impact during usual working hours (6am-6pm) varied in accordance with lifestyle classifications. Specifically, ExWB decreased for all homeworkers but increased significantly for certain hybrid workers. Magnitude of ExWB impact correlated strongly with lifestyle. To understand the significant heterogeneity in ExWB outcomes, a spatial-temporal conceptualisation of working flexibility is developed to explicate the strong yet complex correlations between wellbeing and lifestyles. The implications to post-pandemic “back-to-work” policies are 1) continued expansion of hybrid working optionality, 2) provide wider support for lifestyle adaptation and transitions.”

2. Authors examine the relationship between experiential well-being (ExWB) and remote work, including in their analysis other factors, such as employment characteristics, i.e., hours of work, occupation, or family factors, such as marital status. There is literature that relates parental status (motherhood/parenthood) to remote work and well-being. Authors should justify why they haven’t included in their analysis any indicators for parental status, for example number of children.

Response: Thank you for this important point regarding the lack of ‘parental status’ in our model. While we agree parental status is an important indicator of sociodemographic characteristics, there are several concerns that led us to exclude it from our study. Firstly, there is a significant non-response issue in questions relating to number of children. For instance, in the during-COVID data set there are 227 individuals (out of a total 1,191 samples) who provided no information on whether or not they have children in their household. Excluding these samples would reduce the sample size by roughly 20%, which is likely to lead to identification issues in the model. Moreover, as both childcare and eldercare hours are highly endogenous to the enjoyment of non-paid work, we believe the inclusion of parental status appears outside the scope of this paper. That said, we are interested in looking more closely at the non-paid work activities and analyse the specific impacts of child as well as elderly care in our future workstream. We added the following to the Limitations section of the manuscript to clarify on this point:

“The logistical regression model does not include parental status and household composition as covariates out of concern for endogeneity. While household composition (including parental status) is an important indicator of sociodemographic characteristic, childcare and eldercare hours are highly endogenous to the enjoyment of non-paid work. The relationship between care obligations and ExWB warrant further study before we can address the potential endogeneity. Furthermore, there is a significant non-response issue in questions relating to number of children, affecting 20% of study samples. Inclusion of the covariate would significantly reduce the samples size, likely to lead to identification issues. We believe the inclusion of parental status alone would be problematic and more research is required to differentiate and control for these care obligations.”

3. “The direction and magnitude of pandemic impact on ExWB were heterogenous across time of the day, activity types and lifestyles”. Authors acknowledge the heterogeneous effects of socio-economic factors on ExWB; they should explain better how latent class models help to deal with these heterogeneous effects in the present study.

Response: Thank you for raising the point on heterogenous ExWB changes. The use of latent class models is novel and effective in the sense that it enables a holistic classification of worker, which could hardly be achieved by using conventional, single socio-economic variables. Our empirical analysis showed that the direction and magnitude of wellbeing changes correlate strongly with the lifestyles endogenously identified from the latent class models. We agree with the reviewer that there is untapped potential to use the latent lifestyles as a unified covariate to better control for heterogeneity in the workforce, which is an exciting future workstream for us.

4. I have concerns about how novel is the use of latent class models and time-use survey data. For example:

Lee, Y. Activity Profiles among Older Adults: Latent Class Analysis Using the Korean Time Use Survey. Int. J. Environ. Res. Public Health 2021, 18, 8786. https://doi.org/10.3390/ijerph18168786, uses latent class models along with time-use survey data in a very similar way. Authors should acknowledge relevant to the use of latent class models literature and time-use survey data and motivate better their own contribution.

Response: Thank you for mentioning this relevant paper. We have added a paragraph of detailed review on this paper into our literature review section. From the title and abstract of Lee (2021), the latent class analysis application to time use survey does seem to be a methodological overlap. However, our paper makes several distinct and novel contributions, thus differ from Lee’s methodology in a significant way. 

Firstly, Lee’s paper does not consider the timing (hence the sequence) of activities in the latent class assignment. Our latent classes represent people who engage in similar activities around the similar times of day, as illustrated in our tempogram in Figure 1. Distinguishing the timing and sequence of activities is thus a distinct methodological contribution to the time-use studies. Furthermore, Lee’s paper restrict itself to adults over the age of 65, while we focus on the working population. There is also no modelling of wellbeing nor focus on pandemic-related effects in Lee (2021). Please find the added section in our Literature Review below: 

“Lee (2021) focused on older adults over the age of 65 in the Korean Time Use Survey. Latent Class Analysis was used to identify four activity profiles across two waves in 2014 and 2019. The author aggregates daily activities into eight domains and summed up the time spent in each of these domains. The resulting eight continuous time measurements are then transformed into three-tier ordinal variables, which are used as the basis of classification. This approach focuses on the duration of activities, but ignores the timing and hence sequence of activities. As such, the study is not able to distinguish temporal patterns and sequences in its latent class assignments.”

5. Looking at the number of total observations for each class (Appendix 5), I feel that the authors end up with too many classes and in some cases too few observations for each class. This study would probably benefit from other measures of model of fit and fewer classes or even a different method. These combined can provide to the analysis higher statistical power and more robust results.

Response: Thank you for this comment. Given the lived experience of significant lifestyle changes during the pandemic, it is a key aim of our paper to understand the transition from conventional commuting to more flexible home or hybrid working. Our conceptualisation of working flexibility features interconnected ‘spatial’ (commuting/home/hybrid working) and ‘temporal’ (early start/dispersed/nighttime etc.) dimensions. The combination of these two key dimensions enables a much more nuanced identification of lifestyles. While the number of total classes is high, it is clear from the visualisation that they represent distinct lifestyles that have not yet been explored in flexible working literature. We acknowledge that some classes have fewer samples, they still represent meaningful emerging lifestyles and deserve highlight in our paper. It is a difficult trade-off and we would hope that our nuanced lifestyle classification could be verified by future studies using data of larger sample size.

6. The authors claim that ExWB impact outside of usual working hours (before 6am and after 6pm) was consistently negative for all lifestyles. The choice of these thresholds seems quite arbitrary. Have other authors used such thresholds in the past? Why not for example before 9am and after 5pm? The authors should cite appropriately other authors that have used these thresholds or/and explain better and more the rationale of their choice.

Response: Thank you for raising this important point on the working hour thresholds. Our aim is to include the widest possible range of times that would be considered usual working times. Inherently there are subjective interpretations of when these thresholds should be. Our identification of the 6am-6pm window is data-driven - we see a consistent difference between the periods within and outside the 6am-6pm window across the ten lifestyles identified. While conventionally the 8am/9am-5pm worktime regimes are ingrained in many professional occupations, we take a wider range, as informed by the empirical data, in order to capture the temporal flexibility discussed in our paper.

7. I have concerns about the direction of causality between flexible/remote working and well-being. Some of the data were collected during the pandemic when mental health pressures were exceptionally high. Specifically, I am wondering whether flexible/remote working leads to changes in well-being, or if individuals with better (or worse) well-being are more inclined to work less (or more) flexibly/remotely and travel less (or more). Given the potential for reverse causality, it would be beneficial for this study to include a discussion on the direction of causality between flexible/remote working and ExWB.

While the authors do acknowledge some of the limitations of their study, they do not address potential issues of reverse causality or/and self-selection. So, it is strongly recommended that the authors explore alternative explanations for the observed associations, such as the potential for individuals with better (or worse) well-being to self-select into more (or less) flexible working arrangements.

Response: Thank you for raising the concern regarding causality in our analysis. A similar concern was also raised by the editor. We recognise that our study is focused on associative, rather than causal relationships between time use and wellbeing. This is due to the lack of longitudinal data and the absence of any suitable intervention (treatment) in our data. We have thus followed the suggestions of the editor and removed wording that allude to causality such as “impacts” from applicable places in our manuscript. 

While we agree there are possible instances of circular causality, as our paper is only able to quantify associative relationship, the direction of causality is outside the scope of our consideration. However, if we are able to work with high quality longitudinal data in the future, the potential concerns of circular causality and self-selection would be extremely interesting to our subsequent research. We added the following clarification in the Limitations section of our manuscript:

“Due to the non-interventional data, our study is not able to establish causal relationships – all findings are hence associative. Importan

---

## [Decision Letter · Decision Letter 1]

31 Jan 2024

PONE-D-23-04776R1Remote working and experiential wellbeing: A latent lifestyle perspective using UK Time Use Survey before and during COVID-19PLOS ONE

Dear Dr. Wan,

Thank you for submitting your manuscript to PLOS ONE. After careful consideration, we feel that it has merit but does not fully meet PLOS ONE’s publication criteria as it currently stands. Therefore, we invite you to submit a revised version of the manuscript that addresses the points raised during the review process.

We look forward to receiving your revised manuscript.

Kind regards,

José Alberto Molina

Academic Editor

PLOS ONE

Journal Requirements:

Reviewers' comments:

Reviewer's Responses to Questions

**Comments to the Author**

1. If the authors have adequately addressed your comments raised in a previous round of review and you feel that this manuscript is now acceptable for publication, you may indicate that here to bypass the “Comments to the Author” section, enter your conflict of interest statement in the “Confidential to Editor” section, and submit your "Accept" recommendation.

Reviewer #1: (No Response)

2. Is the manuscript technically sound, and do the data support the conclusions?

Reviewer #1: Yes

3. Has the statistical analysis been performed appropriately and rigorously? 

Reviewer #1: Yes

4. Have the authors made all data underlying the findings in their manuscript fully available?

Reviewer #1: Yes

5. Is the manuscript presented in an intelligible fashion and written in standard English?

Reviewer #1: Yes

6. Review Comments to the Author

Reviewer #1: Referee's Report

Manuscript PONE-D-23-04776R1

'Remote working and experiential wellbeing: A latent lifestyle perspective using UK Time Use Survey before and during COVID-19'

I would like to express my appreciation to the authors for their efforts in addressing the feedback provided. You can find a list of comments below that highlight specific recommendations for further enhancing the paper's potential to offer valuable insights to the field of well-being and remote work. While recognizing the inherent complexities of this research and the limitations posed by available data, I believe that a more detailed explanation of these aspects would significantly strengthen the paper’s overall quality.

With regards to authors’ response to my previous concerns:

1. The length of the abstract is now reduced. However, the authors refer to the time range of 6 am to 6 pm as 'usual working hours' without providing a reference for this threshold later in the paper. Since there is no provided reference for this characterisation (according to authors’ response in comment 6), it would be more accurate to avoid such a characterisation in the abstract and throughout the paper.

2. I appreciate the authors' response and the rationale provided for not including parental status and household composition as covariates in their specification. However, I would like to emphasize the significance of the endogeneity concern in this context. Endogeneity is a bias that can substantially affect the credibility of the study’s findings. As such, it should not be considered 'out of the scope' of the paper. While the authors have stated their reasons for not including these variables, it is crucial that the limitations section of the manuscript more explicitly captures this issue. Specifically, the authors should rephrase their limitation statement to emphasise the potential endogeneity of parental status and household composition in relation to ExWB and remote work and its relevance to their work. They should highlight the complexity of the relationships involved and the need for further research.

3. Could the authors provide a more detailed explanation or illustration of how the latent class models are employed to capture the heterogeneity in ExWB changes among different worker lifestyles? It would be beneficial to understand the specific mechanisms or statistical techniques used within these models that enable a better control of heterogeneity. A more comprehensive description in this regard would greatly enhance the clarity of the methodology and its relevance to the issue of heterogeneous effects, which is a critical aspect of this study.

5. The authors recognize that some classes have fewer samples, which can affect statistical power and the robustness of the results. While they argue that these smaller classes still represent meaningful emerging lifestyles, the issue of statistical power remains a valid concern and should be properly discussed in either the conclusion or limitations section.

6. While I appreciate the authors' response and the rationale provided for choosing the working hour thresholds of 6 am to 6 pm, it is essential to enhance the transparency of this choice for the benefit of readers. I recommend adding a more detailed explanation for this choice in the measurements section of the manuscript.

7. I appreciate the authors' response regarding the concern of causality in the analysis and their efforts to eliminate language implying causality from the manuscript. To ensure consistency throughout the paper and align with the authors' intent to focus on associative relationships, I recommend a careful proofreading of the entire manuscript. This would help identify and appropriately adapt any remaining instances of language suggesting causality. Consistency in terminology will enhance the clarity and accuracy of the paper's purpose.

7. PLOS authors have the option to publish the peer review history of their article (what does this mean?). If published, this will include your full peer review and any attached files.

Reviewer #1: No

---

## [Author Response · Author response to Decision Letter 1]

22 Mar 2024

*Note the response below is attached as Word document for better formatting and readability*

6. Review Comments to the Author

Reviewer #1: Referee's Report

Manuscript PONE-D-23-04776R1

'Remote working and experiential wellbeing: A latent lifestyle perspective using UK Time Use Survey before and during COVID-19'

I would like to express my appreciation to the authors for their efforts in addressing the feedback provided. You can find a list of comments below that highlight specific recommendations for further enhancing the paper's potential to offer valuable insights to the field of well-being and remote work. While recognizing the inherent complexities of this research and the limitations posed by available data, I believe that a more detailed explanation of these aspects would significantly strengthen the paper’s overall quality.

With regards to authors’ response to my previous concerns:

1. The length of the abstract is now reduced. However, the authors refer to the time range of 6 am to 6 pm as 'usual working hours' without providing a reference for this threshold later in the paper. Since there is no provided reference for this characterisation (according to authors’ response in comment 6), it would be more accurate to avoid such a characterisation in the abstract and throughout the paper.

Response: Thank you for this important reminder. We can confirm that all instances of our characterisation of ‘usual working hour’ have been removed from the abstract and throughout the paper. Instead, we conceptualise and differentiate between “daytime activities window (between 6am and 6pm)” and “evening activities window (before 6am and after 6pm)” to clearly bifurcate the 24-hour day into two 12-hour observational windows which are important for our subsequent analysis. The following passage has been added to the manuscript in the Results Section with additional references provided:

“To better understand the heterogeneity observed across the time of the day, we bifurcate the diary day into two 12-hour windows: for daytime activities (between 6am and 6pm) and for evening activities (before 6am and after 6pm). The division recognises the stratifying influence of institutional-organisational time (Scott, 1987). As we use full employment as an inclusion criterium, within-sample must account for collective time schedules – such as those commonly prescribed by the employer during the daytime activities (Flood et al., 2018). We focus primarily on the daytime activities window (between 6am and 6pm) as significant heterogeneity is observed across activities and lifestyles during these hours. In addition to the empirical observation that informed the selection of temporal boundaries for daytime and evening, this window maximises the diversity and richness of daytime activities captured by the granular intra-day temporal data. It expands on the conventional paid work window (such as 9am-5pm), which excludes important activities such as travel and non-paid work which reflect the distinct lifestyles we identified. The 6am and 6pm temporal boundaries result in two comparable observational windows which refines our focus to improve understanding of ExWB and flexible working.”

2. I appreciate the authors' response and the rationale provided for not including parental status and household composition as covariates in their specification. However, I would like to emphasize the significance of the endogeneity concern in this context. Endogeneity is a bias that can substantially affect the credibility of the study’s findings. As such, it should not be considered 'out of the scope' of the paper. While the authors have stated their reasons for not including these variables, it is crucial that the limitations section of the manuscript more explicitly captures this issue. Specifically, the authors should rephrase their limitation statement to emphasise the potential endogeneity of parental status and household composition in relation to ExWB and remote work and its relevance to their work. They should highlight the complexity of the relationships involved and the need for further research.

Response: Thank you to the reviewer for raising the concern regarding endogeneity arising from not controlling for parental status and household composition. We have followed your instruction and revised the limitation statement accordingly. Please see the revised paragraph in the Limitation Section, quoted below:

“The logistical regression model does not include parental status and household composition as covariates. The primary consideration for excluding household composition (including parental status) is a significant non-response issue in the original data where 20% of study samples did not provide the number of children. Inclusion of the covariate would significantly reduce the samples size, leading to identification issues. However, this exclusion may give rise to endogeneity in the logistical regression model. Firstly, childcare and eldercare hours are highly endogenous to the enjoyment of non-paid work, comparing the enjoyment levels during non-paid work activities without explicitly controlling for care duty may risk confounding the coefficients estimated. Secondly, household composition and parental status are important sociodemographic characteristics, the exclusion may lead to incompleteness in the model parameters. Thirdly, without controlling for differences in household composition and parental status weakens the generalisability when comparing ExWB changes before and after the pandemic. Further research is needed to expand the observational unit to the household rather than at the individual level to address the complexities and better address potential endogeneity.”

3. Could the authors provide a more detailed explanation or illustration of how the latent class models are employed to capture the heterogeneity in ExWB changes among different worker lifestyles? It would be beneficial to understand the specific mechanisms or statistical techniques used within these models that enable a better control of heterogeneity. A more comprehensive description in this regard would greatly enhance the clarity of the methodology and its relevance to the issue of heterogeneous effects, which is a critical aspect of this study.

Response: Thank you for raising the need for further explaining the LCA model specifications. While the key considerations for model choices are summarised at the start of the Statistical Analysis Section:

“Latent class analysis (LCA) is used to identify nuanced lifestyles within each working mode group (homeworkers, commuters, and hybrid workers). LCA is a generalised structural equation model used to identify a latent (unobserved) categorical variable based on a set of observed categorical variables (Sinha et al., 2021). It is used to find groups with similar features or to decipher distinct patterns in multivariate data through maximizing cross-group variance and minimising within-group variance. The goal is to simplify and visualise complex and high-dimensional data in an unsupervised manner. Identification of LCA classes is particularly useful in largescale observational studies and helps contextualise unobserved heterogeneity. In particular, the relatively homogenous groups identified as a result of the LCA models have distinctive spatial and temporal features representing interpretable lifestyles. The identification of lifestyle delineates heterogenous subgroups of the sample that would have otherwise aggregated as a homogenous sample, mitigating the risk of inadvertently averaging out important heterogeneities in these unidentified latent subgroups. For key formulae and technical specifications see S5 Appendix.”

Furthermore, we agree with the reviewer that given the importance of the LCA methodological choice, the technique should be explained in greater detail. We thus added the key equations and parameters in the Supplemental Materials:

“S5 Appendix: Key formulae for LCA 

P_jkt= P_t^X P_jt^AX P_kt^BX

(Equation 1)

X is identifiable as a latent variable with T=1…t classes so that P_t^Xis the probability of being in latent class t. P_jt^AX is the probability of being in category j of variable A conditional on being in latent class t. P_kt^BXis the probability of being in category k of variable B conditional on being in latent class t. P_jkt is the joint probability of being in category j, category k, and class t.

The responses of individual i on variables A and B are denoted each as Y_ijk, which is represented in the following section as a vector Y_i, such that the latent class model is defined as:

P(Y_i )=∑_(t=1)^T▒〖P(X=t)P(Y_i |X=t)〗

(Equation 2)

Where P(X=t) denote the class proportions. Applying a simple Bayesian formula to obtain the probability of belonging to latent class t given the observed response vector Y_i, the posterior membership probability is obtained from (Weller et al., 2020):

P(X=t│Y_i )=(P(X=t)P(Y_i |X=t))/P(Y_i ) 

(Equation 3)

Further defining the assigned latent class membership of individual i as W_i. Adopting a proportional classification, which yields a soft partitioning with boundary s for assigning individuals to latent class t denoted as W_it=P(W_i=t│Y_i )=P(X=t│Y_i ). Allowing the classification error to be defined in terms of probability of the estimated value conditional on the true value:

P(W=s│X=t)=(∑_(i=1)^N▒〖P(X=t│Y_i )P(W=s│Y_i ) 〗)/(P(X=t))=(∑_(i=1)^N▒〖W_it W_is 〗)/(P(X=t))

(Equation 4)

24 hourly categorical variables indicating activity type were used as the observed categorical variables in the models, together with categorical socioeconomic covariates including income quantile, education attainment, and occupation classification. Following the three-step model (Bolck et al., 2004), incorporating covariate with response vector Z_i into the latent class model allows the covariates effects to be simultaneously estimated with the parameters defining the class-specific item distributions such that:

P(W=s|Z_i )=∑_(t=1)^T▒〖P(X=t|Z_i)P(W=s│X=t) 〗

(Equation 5)

A multinomial logistical regression is used to parametrise P(X=t|Z_i). Using a maximum log-likelihood function, the model parameters above are estimated with:

LogL=∑_(i=1)^N▒〖log⁡〖P(Y_i│Z_i )〗=∑_(i=1)^N▒log⁡∑_(t=1)^T▒〖P(X=t|Z_i)P(W=s│X=t) 〗 〗

(Equation 6)

To ensure the global rather than local maximum is identified in the maximum likelihood estimation, Newton-Raphson stepping mechanism guarantees convergence if the initial estimate x_0 are close enough to the true estimates (Smith, 1998), as shown in the formula:

x_(n+1)=x_n-(f(x_n))/(f'(x_n))

(Equation 7)

To narrow the initial estimates to aid model convergence under Newton method, we apply a two-step integration process in the generalised linear mixed model used for LCA: Laplacian approximation was used in the first round of integration, the resulting matrix was then used as the starting value for the more accurate mean-variance adaptive Gauss-Hermite quadrature integration method (Rabe-Hesketh et al., 2002):

∫_(-∞)^∞▒〖L(β_0,β_1;Y_i |X=t)P(X=t|Z_i)dx〗

(Equation 8)

LogL(β_0,β_1;Y_i│X=t)=Y_i (β_0+β_1 x)-log⁡〖(1+e^(β_0+β_1 x))〗

(Equation 9)

One to six class models are iteratively attempted, the final class size within each working mode was determined using the Akaike's information criterion (AIC) and Bayesian information criterion (BIC) values, average latent class probabilities, and substantive interpretation of the classes identified:

AIC=-2 log⁡L(θ ^ )+2k

(Equation 10)

BIC=-2 log⁡L(θ ^ )+k log⁡n

(Equation 11)

Where θ is the vector of model parameters and k is the number of estimated parameters. L(θ ^ ) is the candidate model when evaluated at the maximum likelihood estimate of θ. LCA was conducted using pooled samples from both pre- and during COVID-19. After LCA, the samples were then separated into pre- and during COVID-19 groups according to the survey date.”

*Note: please kindly note there was no Comment 4 from the reviewer in the Decision Letter, possibly due to numbering error (i.e. Comment 3 above immediate preceded Comment 5 below). We thus answer Comment 5 below without re-numbering the comments for consistency.

5. The authors recognize that some classes have fewer samples, which can affect statistical power and the robustness of the results. While they argue that these smaller classes still represent meaningful emerging lifestyles, the issue of statistical power remains a valid concern and should be properly discussed in either the conclusion or limitations section.

Response: Thank the reviewer for raising the concern regarding the uneven sizes of latent classes. We acknowledge this limitation and have added the following discussion to the Limitation Section:

“A minority of lifestyles identified from LCA models have relatively small class sizes, in particular two of the hybrid lifestyles during the pandemic (LH and MH have 30 and 36 individuals respectively). Compared to lifestyles with larger class size, the ExWB averages produced may be subject to the effects of outliers. While the logistical regression produced highly statistically significant estimates for the smaller LH and MH lifestyles during the pandemic (>99% significance), it does not rule out possible limitation affecting the external validity of these estimates due to smaller sample sizes. This limitation resulted from a confluence of a minority of workers engaging in flexible working lifestyles such as hybrid lifestyles, as well as the data restriction of fewer individuals in the during-COVID dataset. As hybrid working becomes more prevalent and more post-pandemic data is gathered, it would be useful to verify the external validity of the findings for the smaller LCA classes of hybrid workers in future studies.”

6. While I appreciate the authors' response and the rationale provided for choosing the working hour thresholds of 6 am to 6 pm, it is essential to enhance the transparency of this choice for the benefit of readers. I recommend adding a more detailed explanation for this choice in the measurements section of the manuscript.

Response: Thank you for the cogent comment. We agree and have revised the characterisation following the reviewer’s Comment 1. We appreciate the reviewer’s concern regarding the need to signpost this methodological choice earlier in the manuscript, we added the following passage to the Measurement Section:

“Recognising the rigidity of certain working arrangement (e.g. the eight-hour working day that has largely persisted since the industrial era) and the need to capture lifestyle differences before and during COVID-19, the dairy day for each respondent is bifurcated into a “daytime activities window” (between 6am and 6pm) and a “evening activities window” (before 6am and after 6pm). The 12 consecutive categorical hourly variables thus fall into each observational window in a mutually exclusive manner, with 6am and 6pm as the boundaries. The rationale for these temporal boundaries is based on empirical observations (see details in the Results section).”

7. I appreciate the authors' response regarding the concern of causality in the analysis and their efforts to eliminate language implying causality from the manuscript. To ensure consistency throughout the paper and align with the authors' intent to focus on associative relationships, I recommend a careful proofreading of the entire manuscript. This would help identify and appropriately adapt any remaining instances of language suggesting causality. Consistency in terminology will enhance the clarity and accuracy of the paper's purpose.

Response: Thank the reviewer for raising the importance of consistency regarding the use of associative relationships, we confirm that this issue has now been addressed throughout the manuscript. In particular, we confirm that any references to causal language such as “impacts” or “effects” have been replaced w

---

## [Decision Letter · Decision Letter 2]

17 Apr 2024

PONE-D-23-04776R2Remote working and experiential wellbeing: A latent lifestyle perspective using UK Time Use Survey before and during COVID-19PLOS ONE

Dear Dr. Wan,

Thank you for submitting your manuscript to PLOS ONE. After careful consideration, we feel that it has merit but does not fully meet PLOS ONE’s publication criteria as it currently stands. Therefore, we invite you to submit a revised version of the manuscript that addresses the points raised during the review process.

We look forward to receiving your revised manuscript.

Kind regards,

José Alberto Molina

Academic Editor

PLOS ONE

Journal Requirements:

Reviewers' comments:

Reviewer's Responses to Questions

**Comments to the Author**

1. If the authors have adequately addressed your comments raised in a previous round of review and you feel that this manuscript is now acceptable for publication, you may indicate that here to bypass the “Comments to the Author” section, enter your conflict of interest statement in the “Confidential to Editor” section, and submit your "Accept" recommendation.

Reviewer #1: All comments have been addressed

Reviewer #3: (No Response)

2. Is the manuscript technically sound, and do the data support the conclusions?

Reviewer #1: Yes

Reviewer #3: Yes

3. Has the statistical analysis been performed appropriately and rigorously? 

Reviewer #1: Yes

Reviewer #3: Yes

4. Have the authors made all data underlying the findings in their manuscript fully available?

Reviewer #1: Yes

Reviewer #3: Yes

5. Is the manuscript presented in an intelligible fashion and written in standard English?

Reviewer #1: Yes

Reviewer #3: Yes

6. Review Comments to the Author

Reviewer #1: No further comments. There are still some minor grammatical errors and typos. For example, 'Limitation' section should be 'Limitations'. I recommend a full proofreading of the paper.

Reviewer #3: The current version of the paper is quite well. Two final comments:

1 If the UKTUS is cross-sectional, and additional waves to the 2015 wave are “population-representative”, I wonder whether individuals' assessment of well-being at work might be affected by whether or not individuals remain employed. Let me explain my point of view. If many people lost their jobs, or are on flexible time schemes not working, during the lockdowns, individuals who remain employed during COVID may feel better off compared to similar individuals Pre-COVID, simply because in "bad times" they are still employed. This "possible" source of bias may occur regardless of lifestyle, and worker mode. The authors may find this hypothesis relevant.

2 In the section "Time use and ExWB during the evening activities window", first sentence. "ExWB decreased uniformly across all lifestyles within the evening activities". I don't quite understand the sentence. The later in the evening the less satisfied with activities or only when working? or is there some gradation in lifestyles? I may have misunderstood the process.

7. PLOS authors have the option to publish the peer review history of their article (what does this mean?). If published, this will include your full peer review and any attached files.

Reviewer #1: No

Reviewer #3: No

---

## [Author Response · Author response to Decision Letter 2]

22 May 2024

Reviewer #1: No further comments. There are still some minor grammatical errors and typos. For example, 'Limitation' section should be 'Limitations'. I recommend a full proofreading of the paper.

Response: Thank you to the reviewer for the reminder on grammatical accuracy. We confirm a full proofreading has been undertaken by both authors. We made the following changes in the proofreading process:

1. Section head “Limitation” was revised as “Limitations”;

2. In Abstract, “It was found that” was deleted for passive voice;

3. In Introduction, “and social scarring” was deleted for better readability;

4. In Introduction, comma added for readability: “… the lifestyle perspective and the temporal dimension combined are new, thus contribute to our understanding of the association between flexible/remote working and wellbeing.”;

5. In Literature Review, “as well” was removed: “…poses a large burden for the respondents” to improve readability;

6. In Literature Review, “taken place” was removed;

7. In Literature Review, the acronym “ONS” has been expanded to “Office for National Statistics”;

8. In Data and Methods, “endogenous” was removed and “mode” was changed to model as a correction for spelling in the sentence “Prior to the identification of lifestyles using the latent class models…”;

9. In Data and Methods, “they” was clarified as “workers”;

10. In Results, Table 1, “demographies” was changed to “samples”; “working from home” was changed to “homeworking”; “lockdown” was changed to “COVID-19”; “spends the most time” was changed to “spends the longest duration”;

11. In Results, “homeworker groups” was changed to “homeworking lifestyles”;

12. In Results, “each reflecting” was changed to “respectively for”;

13. In Results, “turn to negative” was changed to “turn negative”;

14. In Results, two instances of “effect” was changed to “changes” to avoid causal insinuation;

15. In Conclusion, “It was found that” was deleted for passive voice.

All changes are recorded in the revised manuscript using tracked changes.

Reviewer #3: The current version of the paper is quite well. Two final comments:

1 If the UKTUS is cross-sectional, and additional waves to the 2015 wave are “population-representative”, I wonder whether individuals' assessment of well-being at work might be affected by whether or not individuals remain employed. Let me explain my point of view. If many people lost their jobs, or are on flexible time schemes not working, during the lockdowns, individuals who remain employed during COVID may feel better off compared to similar individuals Pre-COVID, simply because in "bad times" they are still employed. This "possible" source of bias may occur regardless of lifestyle, and worker mode. The authors may find this hypothesis relevant.

Response: Thank you to the reviewer for the cogent comment regarding the possible upward bias in ExWB for those who remain employed during the COVID pandemic. The reviewer posed a valid and intriguing hypothesis – given the broader social context of mass layoffs or furlough schemes, it is possible that individuals who keep their employment receive a ExWB increase. We refer this possible ExWB increase for those remaining employed as ‘background effect' in the following discussion.

Firstly, our samples are restricted to full time workers in both UKTUS (2015) and the following waves. It is, however, not possible to track the change of employment status over time due to the cross-sectional nature of our samples. We observed highly heterogenous changes in ExWB (both in direction and magnitude) during the pandemic – particularly during hours with high concentrations of workers engaging in paid work. To investigate whether, and if so, how the ‘background effect’ would influence our findings, it is important to refine the hypothesis by considering whether the effect is uniform among those remained full-time employed. We will discuss two cases below. 

If the background effect is uniform among our full-time worker samples, it may introduce a uniform bias (in terms of the lifestyle groups) to our logistic regression model estimates. However, the core finding on heterogenous changes in ExWB (both in direction and magnitude) remains valid, as the uniform bias would not alter the relative differences between lifestyles (the identification of which is independent of ExWB measures). 

If the background effect is non-uniform among our samples, such effect, in theory, has already been accounted for in our empirical models as part of the observed heterogeneity. In other words, the possible existence of background effect would support our core finding, though likely to derive from a different source. We agree with the reviewer that it would be useful to investigate and isolate the various sources of the heterogenous effects on ExWB, but such a task would require a causal framework which falls out of the scope of the paper. We have made considerable efforts in the previous revisions to clarify that our study is not causal but based on association. 

Based on the above analysis, we appreciate that the background effect as hypothesised by the reviewer helps us to gain a deeper understanding of the observed heterogenous effects on ExWB. The possible existence of the background effect does raise a potential issue of biased estimates in our logistic models, but overall it does not alter our core findings. On the contrary, it may corroborate our findings if the background effect itself was heterogenous across employed workers. We would like to investigate the possible linkage between broader labour market changes and ExWB in future studies. Accordingly, we added the following passage to our Limitations section to clarify the potential bias: 

“The lack of longitudinal data limits our ability to control for employment change across waves, which we addressed by restricting samples to full-time workers at each data cross-section for comparability. On an aggregate level, the logistical regression model results cannot capture changes in the labour market and the associated background effects on ExWB. Specifically, the background effects may be linked to an upward bias in ExWB at the start of the pandemic, as our samples were fully employed despite the labour market contraction. If such background effects were present and uniformly impacted samples during COVID-19, the relative differences in ExWB across lifestyles would remain constant. On the other hand, non-uniform background effects may have contributed to, and help explain, the observed heterogeneity between lifestyles. In either case, background effects associated with labour market changes would not contradict our core findings. Longitudinal data would be required to identify the presence of such effects, and to delineate the sources of heterogeneity.”

2 In the section "Time use and ExWB during the evening activities window", first sentence. "ExWB decreased uniformly across all lifestyles within the evening activities". I don't quite understand the sentence. The later in the evening the less satisfied with activities or only when working? or is there some gradation in lifestyles? I may have misunderstood the process.

Response: Thank you for the comment. In the sentence highlighted by the reviewer, we are comparing the evening activities window pre- and during COVID-19, rather than the trend within the evening activities window itself (i.e. early evening versus later in the evening). Empirical results show a decline of ExWB in the evening activities window during COVID-19, compared against the pre-COVID level, and the reduction is uniform across all lifestyles. We do not observe a significant temporal gradation in ExWB within the evening activities across lifestyles. We have revised this sentence for better clarity. Please see the revision below:

“During COVID, ExWB declined for all lifestyles within the evening activities window when most people engaged in leisure or personal activities such as sleeping.”

---

## [Decision Letter · Decision Letter 3]

24 May 2024

Remote working and experiential wellbeing: A latent lifestyle perspective using UK Time Use Survey before and during COVID-19

PONE-D-23-04776R3

Dear Dr. Wan,

We’re pleased to inform you that your manuscript has been judged scientifically suitable for publication and will be formally accepted for publication once it meets all outstanding technical requirements.

Kind regards,

José Alberto Molina

Academic Editor

PLOS ONE

Additional Editor Comments (optional):

Reviewers' comments:

Reviewer's Responses to Questions

**Comments to the Author**

1. If the authors have adequately addressed your comments raised in a previous round of review and you feel that this manuscript is now acceptable for publication, you may indicate that here to bypass the “Comments to the Author” section, enter your conflict of interest statement in the “Confidential to Editor” section, and submit your "Accept" recommendation.

Reviewer #3: All comments have been addressed

2. Is the manuscript technically sound, and do the data support the conclusions?

Reviewer #3: Yes

3. Has the statistical analysis been performed appropriately and rigorously? 

Reviewer #3: Yes

4. Have the authors made all data underlying the findings in their manuscript fully available?

Reviewer #3: No

5. Is the manuscript presented in an intelligible fashion and written in standard English?

Reviewer #3: Yes

6. Review Comments to the Author

Reviewer #3: It's ok. The authors have sastisfactorily addressed I raised. The current version is ok for me and now the editor should decide on it

7. PLOS authors have the option to publish the peer review history of their article (what does this mean?). If published, this will include your full peer review and any attached files.

Reviewer #3: No

---

## [Editor Report · Acceptance letter]

16 Jul 2024

PONE-D-23-04776R3 

PLOS ONE

Dear Dr. Wan, 

I'm pleased to inform you that your manuscript has been deemed suitable for publication in PLOS ONE. Congratulations! Your manuscript is now being handed over to our production team.

Kind regards, 

on behalf of

Professor José Alberto Molina 

Academic Editor

PLOS ONE